# Neighborhood and Depressive Symptoms in Older Adults Living in Rural and Urban Regions in South Korea

**DOI:** 10.3390/healthcare11040476

**Published:** 2023-02-07

**Authors:** Seon Kim, Sunghwan Cho, Matthew R. Morgan

**Affiliations:** School of Social Work, Virginia Commonwealth University, Richmond, VA 23284, USA

**Keywords:** older adults, depressive symptoms, development of neighborhood, rural, urban

## Abstract

Neighborhoods have a significant impact on depressive symptoms in older adults. In response to the increasing depression of older adults in Korea, this study aims to identify the relationship between perceived and objective neighborhood characteristics in depressive symptoms and find differences between rural and urban areas. We used a National survey collected in 2020 of 10,097 Korean older adults aged 65 and older. We also utilized Korean administration data for identifying the objective neighborhood characteristics. Multilevel modeling results indicated that depressive symptoms decreased when older adults perceived their housing condition (b = −0.04, *p* < 0.001), their interaction with neighbors (b = −0.02, *p* < 0.001), and overall neighborhood environment (b = −0.02, *p* < 0.001) positively. Among the objective neighborhood characteristics, only nursing homes (b = 0.09, *p* < 0.05) were related to depressive symptoms of older adults living in urban areas. For older adults living in rural areas, the number of social workers (b = −0.03, *p* < 0.001), the number of senior centers (b = −0.45, *p* < 0.001), and nursing home (b = −3.30, *p* < 0.001) in the neighborhood were negatively associated with depressive symptoms. This study found that rural and urban areas have different neighborhood characteristics related to older adults’ depressive symptoms in South Korea. This study encourages policymakers to consider neighborhood characteristics to improve the mental health of older adults.

## 1. Introduction

Depressive symptoms among older adults are becoming a serious social problem in South Korea. More than 160,000 new cases of depressive symptoms in older adults occur every year [1]. Older adults are more likely to experience depressive symptoms than other population groups because they are socially isolated and more likely to experience economic hardship and health problems [2,3]. Depressive symptoms are considered early-onset symptoms of dementia [4,5] and are also related to suicide [6] and chronic health conditions [7]. Depressive symptoms of older adults can also exacerbate the informal caregiver burden [8]. As depressive symptoms affect older individuals and caregivers, it is essential to identify factors related to depressive symptoms in a policy focused on older adults.

Neighborhood environment has a more significant effect on older adults compared to younger adults [9]. Since older adults are more likely to experience reduced physical functioning and have limited mobility, they typically stay in the same area longer than younger adults [2,10]. Depressive symptoms in older adults differ depending on how they interact with their neighborhoods [11,12]. Lawton and Nahemow [12] argued, using the competence-environmental press model, that the interaction between an individual’s competencies and the environment can determine a health outcome, including mental health. For example, a lack of medical professionals increases the risk of depression in older adults who have medical problems [13]. Also, older adults with mobility issues are at higher risk of depressive symptoms when facing environmental barriers such as limited transportation and living a long distance from family and friends [13,14].

There is a lack of research on the impact of neighborhoods on depressive symptoms among older adults [14]. Moreover, most of the neighborhood studies on health outcomes have been conducted in public health, and the social work perspectives have been neglected [15]. This study intends to present additional evidence on the relationship between neighborhood characteristics and depressive symptoms among older adults in South Korea and to suggest implications for social work policy based on the results.

### 1.1. Key Neighborhood Characteristics

Depressive symptoms correlate with the subjectively evaluated neighborhood characteristics [16]. One of the most common measures used to evaluate neighborhood characteristics is the age-friendly cities (AFC) framework developed by the World Health Organization [17]. The age-friendly features proposed by WHO include: housing, outdoor spaces and buildings, transportation, civic participation and employment, social participation, respect and social inclusion, communication and information, and community and health services. According to Wang and colleagues [18], depressive symptoms were alleviated when older adults positively perceived outdoor space, transportation, housing, leisure facilities, and access to health facilities. Kim and Youm [19] also said that neighborhood factors perceived by older adults, such as housing stability and fear of neighbors, are highly related to depressive symptoms.

Objective neighborhood characteristics also relate to the depressive symptoms of older adults. Demographic composition, including the proportion of the older population, and the proportion of people living in poverty, can make a difference in providing health and social services [20,21,22]. This imbalance in community-based healthcare resources is more likely to be experienced by people living in isolated and underserved areas and can lead to psychological distress [23]. Wandersman and Nation [24] also suggested the structural characteristics model that describes the relationship between neighborhoods and mental health. In this model, the demographic characteristics of a population, such as the poverty rate and proportion of race/ethnicity in the neighborhood, are referred to as structural characteristics [24]. The model explained that variation in neighborhood characteristics creates psychological stress and social organization, which affect mental health outcomes [24].

Depressive symptoms are more severe among older adults in rural areas [4,18]. In Korea, rapid industrialization has resulted in the concentration of population and social service-related facilities in large cities, and older adults in rural areas have been left out [21]. Rural areas have a high proportion of older adults and often lack well-developed healthcare facilities [4]. On the other hand, urban areas reported relatively low depressive symptoms due to greater access to medical and social services [4]. In Korea, urban and rural environments are markedly different [21]. There is a dearth of studies among Korean older adults examining the relationship between depressive symptoms and neighborhoods and comparing urban and rural areas.

### 1.2. Present Study

Few studies have simultaneously considered the perceived and objective neighborhood characteristics of older adults’ depressive symptoms. Also, there are apparent differences between rural and urban characteristics. This study explores perceived and objective neighborhood characteristics and differences between rural and urban areas related to depression in Korean older adults. As shown in the conceptual model Figure 1, the research questions of this study are as follows: (1) What are perceived neighborhood characteristics associated with depressive symptoms in older adults? (2) What are objective neighborhood characteristics associated with depressive symptoms in older adults? (3) Are there different neighborhood factors associated with depressive symptoms in rural and urban older adults?

## 2. Methods

### 2.1. Data and Sample

The study used data from the 2020 National Survey of Older Koreans and Korean administrative data [25]. Korea Institute for Health and Social Affairs [2] conducted the 2020 National Survey of Older Koreans data every three years from 2008 based on Article 5 of the Older Adult Welfare Act. This Act aims to contribute to promoting the health and welfare of older adults by maintaining mental and physical health through appropriate treatment and recovery. The survey proceeds with the following objectives: (1) it provides reference data to identify Korean older adults’ living conditions, characteristics, and needs to prepare welfare policies to improve their quality of life; (2) to identify the changing characteristics of older adults by using time-series data.

10,097 Korean older adults aged 65 and over participated in the fifth survey that was conducted on September 14 to 20 November 2020. We used the entire sample for our analysis. The participants answered the questions based on their experience in 2019 except for depressive symptoms. The in-person interview was conducted from September 14 to 20 November 2020, using the Tablet-PC Assisted Personal Interview (TAPI) method by 169 interviewers who received training based on the questionnaire designed by the research team [2]. Using the 2018 Population and Housing Census survey, the whole country was stratified into eight metropolitan cities and nine provinces [2]. Eight metropolitan cities include Seoul, Busan, Daegu, Incheon, Gwangju, Daejeon, Ulsan, Sejong, and nine provinces including Gyeonggi, Kangwon, Chungbuk, Chungnam, Chonbuk, Chonnam, Gyeongbuk, Gyeongnam, and Jeju. We linked this in-person survey data with Korean administrative data that measure neighborhood-level characteristics (e.g., the proportion of older adults, and the number of social workers) as of December 2019. Based on previous neighborhood studies [21,25], this study obtained neighborhood-level objective information from the Korean Statistics Information Service [26].

### 2.2. Measures

#### 2.2.1. Depressive Symptoms

We used the Korean version of the Short form of the Geriatric Depression Scale (SGDS-K) to identify Korean older adults’ depressive symptoms. The SGDS-K is a self-rated 15-item scale with a range of 0–15 points; a cutoff score of 8 points or higher indicates the presence of depressive symptoms [2]. Some questions were reverse-coded to create the sum value. Some example questions from the instrument include: “Choose the best answer for how you have felt over the past week;” “Are you basically satisfied with your life?” and “have you dropped many of your activities and interests?” The SGDS-K had good reliability and validity for assessing Korean older adults’ depression [27]. This measurement also showed good reliability (Cronbach’s α = 0.850) in this study.

#### 2.2.2. Perceived Neighborhood Characteristics

Eight neighborhood factors measured satisfaction with housing, distance from the facilities, public transportation, green space sufficiency, public safety, distance from the family members, interaction with neighbors, and overall community environment. To assess these variables, respondents were asked the general question “how satisfied are you with the community environment in which you live? They then marked their responses for each of the eight factors. Respondents rated their satisfaction level on a five-point Likert scale (5 = excellent, and 1 = poor).

#### 2.2.3. Objective Neighborhood Characteristics

We included 8 variables as objective neighborhood characteristics based on the previous literature [20,21,22] and older adults’ social services (i.e., senior centers, nursing homes, and home care services) and facilities surveyed by all local governments. We included the number of public social work administrators (who are public officials in charge of social work in municipalities), social workers, senior centers, nursing homes, and home care services per 1000 older adults. The proportion of older adults was the number of older adults to the total number of the population in the neighborhood. To understand the level of socioeconomic status in each area, we added two variables: the region’s per capita income and financial independence rate. The region’s per capita income was measured as a continuous variable, and the unit for income was 1000 South Korean Won (KRW). Financial independence was the percentage of budgets not dependent on the central government in each municipality, and it was calculated by “state tax + non-tax income—state debt)/general accounting revenue ∗ 100”.

#### 2.2.4. Rural/Urban

In Korea, an area with more than 50,000 residents is defined as urban [28]. Administrative districts in Korea are currently divided into ‘metropolitan cities’, ‘provinces’, ‘cities’, ‘Eup’, ‘Myeon’, and ‘Dong’ considering the population size [29].‘Eup’ is similar to the unit of town in the US. ‘Myeon’ (township) has a less dense population than ‘Eup’ and represents the rural areas of the more populated division “Goon” (County). The dong (neighborhood) is the smallest administrative division of districts in urban areas [29]. ‘Eup’ and ‘Myeon’ are defined as rural areas, and ‘Dong’ are defined as urban areas in this study.

#### 2.2.5. Covariates

Based on previous studies, we included several individual factors as covariates. Age, household size, and the household’s annual income were derived from the actual numbers reported by respondents. Gender was measured in two ways by biological sex (1 = male, 2 = female). We measured marital status based on the existence of a partner (0 = single, 1 = couple). Respondent’s education ranged from 1 (no education) to 7 (a bachelor’s degree or higher). Regarding employment status, the participant could answer “yes” if they had been paid for working more than 1 h in the past week or “no” if they have not. For self-rated health (SRH), respondents rated their overall health status on a five-point Likert scale (5 = excellent, 1 = poor). Regarding the disability, the question was “are you a person with a disability?” To objectively understand the health status of older adults, the respondent answered the number of chronic diseases suffered over three months.

#### 2.2.6. Analysis Plan

We conducted descriptive, bivariate, and multilevel modeling (MLM) analyses. Descriptive findings presented the basic information of each variable for the entire sample and compared the neighborhood characteristics with the rural and urban areas using t-test and ANOVA. For the MLM analysis, we followed the approach by Garson [30] to build the MLM and previous studies regarding neighborhood environment [25,31,32].

First, a null model was fit to be determined by neighborhood group variables which included eight metropolitan cities and nine provinces (e.g., Seoul, Busan, Incheon, Gyeonggi). We found a 4 percent variance in depression (ICC = 0.04), which was generally perceived as a low value for performing a multilevel model. However, a low ICC was common in observational studies [33] and other studies related to neighborhood and health outcomes [31,34]. Model 1 included individual-level factors, Model 2 included perceived neighborhood characteristics after applying individual-level factors, and Model 3 added objective neighborhood characteristics. This study performed MLM separately in model 4 to identify relationships between neighborhood characteristics and rural (4a) and urban (4b) areas. We used likelihood estimation for each model to ensure the comparability between different models through fit indices: Akaike information criterion [AIC] and Bayesian information criterion [BIC] [35,36]. We used SPSS 27.0 for descriptives, bivariate and data cleaning, and Stata 14 for multilevel modeling.

## 3. Results

### 3.1. Descriptive/Bivariate

Table 1 presents descriptive findings. The mean score of depressive symptoms in the entire sample was 3.4 (SD = 3.4), and depressive symptoms were statistically higher in rural areas (M = 3.6, SD = 3.4) than in urban areas (M = 3.3, SD = 3.3). Regarding individual-level factors, the average age was 73.6 (SD = 6.3), most of the sample was female (60.0%), 58.7% of older adults had partners, and the mean value of household size was 1.9 (SD = 0.8). The average education level of the respondents was 3.8 (SD = 1.2) out of 7, which was between elementary school and middle school. The average monthly income was 1,489,300 KRW which was approximately 1203 US dollars. 35.9% of the respondents were employed. The mean score of self-rated health was 3.3 (SD = 0.9) out of 5, which was between fair and good. 4.5% of the respondents have a disability. The average number of chronic diseases was 1.9 (SD = 1.5). Among the individual factors, age, household size, education, income, employment, self-rated health, and disability were statistically different between rural and urban areas.

Regarding the perceived neighborhood characteristics, the sample showed a satisfaction between 3.5 and 3.8 out of 5 in the entire sample. All variables showed better satisfaction levels in urban areas. Among them, housing (M = 3.8, SD = 0.7), distance from facilities (M = 3.8, SD = 0.7), public transportation (M = 3.9, SD = 0.7), public safety (M = 3.8, SD = 0.7), distance from the family members (M = 3.6, SD = 0.9), interaction with neighbors (M = 3.8, SD = 0.7), overall community environment (M = 3.7, SD = 0.7) were statistically higher in urban areas than rural areas.

The average number of public social work administrators was 3.2 (SD = 0.0), and social workers were 75.2 (SD = 0.2) per thousand older adults. The average number of senior centers per thousand older adults was 3.3 (SD = 0.5), which was higher than nursing homes (M = 0.7, SD = 0.0), and home care services (M = 0.6, SD = 0.0). The average region’s per capita income was 3,579,500 won (SD = 70.0), and the financial independence rate was 43.5% (SD = 0.3). The average proportion of older adults was 16.2 (SD = 3.3). The mean value of the proportion of older adults (M = 18.5, SD = 3.0) and public social work administrators (M = 3.3, SD = 0.5) was statistically higher in the rural area, social workers (M = 76.7, SD = 20.1), and region’s per capita income (M = 36,249.2, SD = 7881.6), and financial independence rate (M = 48.9%, SD = 16.7) were higher in the urban area.

### 3.2. Multilevel Models

Table 2 presents the result of multilevel models in the full sample and by rural/urban areas. Several individual-level factors were associated with depressive symptoms, including gender (b = −0.01, *p* < 0.01); marital status (b = −0.03, *p* < 0.001); education (b = −0.01, *p* < 0.001); income (b = −2.305 × 10^−6^, *p* < 0.001); employment (b = −0.02, *p* < 0.001); self-rated health (b = −0.07, *p* < 0.001); disability (b = 0.05, *p* < 0.001); chronic diseases (b = 0.02, *p* < 0.001) in the final model 1. This indicates that females were more likely to have depressive symptoms than their male counterparts. In addition, older adults who don’t have a partner, have lower education levels, have lower income, are unemployed, have poorer self-rated health, and have more chronic diseases, are more likely to have depressive symptoms. Individual characteristics showed a similar tendency in urban and rural areas, but as age and household size increased, the depressive symptoms decreased statistically in rural areas. The higher education level significantly reduced depressive symptoms in urban areas while it was not statistically related to depressive symptoms increased in rural areas. The education level is positively related to depressive symptoms in urban areas while it was not statistically related to depressive symptoms in rural areas. 

After applying the individual factors (model 1), model 2 showed that several perceived neighborhood characteristics were significantly related to depressive symptoms among older adults. First, satisfaction with housing (b = −0.04, *p* < 0.001), interaction with neighbors (b = −0.02, *p* < 0.001), and overall community environment (b = −0.02, *p* < 0.001) were negatively associated with depressive symptoms. Public transportation (b = 0.01, *p* < 0.01) was positively related to depressive symptoms. These results showed that older adults were less likely to be depressed when satisfied with housing, interaction with neighbors, and the overall community environment. Model 2 showed better-fit indices than Model 1 (AIC = −4457.98, BIC = −4025.35).

In model 3, the number of senior centers (b = 0.01, *p* < 0.05) and nursing homes (b = 0.09, *p* < 0.01) were positively related to depressive symptoms. Thus, older adults who use senior centers and nursing homes were more likely to have depressive symptoms. AIC value became a little better than the previous model. However, BIC increased when we added objective neighborhood characteristics. Since both values did not show a better fit at the same time, objective neighborhood characteristics may not show a statistical change in the entire sample.

We found different results between rural and urban areas (Models 4a and 4b). All objective neighborhood characteristics were related to depressive symptoms in rural areas. The number of public social work administrators (b = 3.10, *p* < 0.001), home care service (b = 0.68, *p* < 0.001), and the proportion of older adults (b = 0.07, *p* < 0.01) were positively related to depressive symptoms in older adults living in the rural area. The number of senior centers (b = −0.45, *p* < 0.001), nursing homes (b = −3.30, *p* < 0.001), and the region’s per capita income (b = −0.00, *p* < 0.001) were negatively related to depressive symptoms. However, only nursing homes (b = 0.09, *p* < 0.05) were positively related to depressive symptoms in urban areas.

Among the perceived neighborhood characteristics, satisfaction with distance from the facilities (b = 0.01, *p* < 0.05) and public transportation (b = 0.01, *p* < 0.05) were positively related to depressive symptoms in rural areas, while satisfaction with interaction with neighbors (b = −0.02, *p* < 0.001) and overall community environments (b = −0.02, *p* < 0.001) were negatively related to depressive symptoms in the urban area. Satisfaction with housing was negatively related to depressive symptoms in both areas.

## 4. Discussion and Implications

This study contributed to the literature by testing the structural characteristics model with older Korean adults using a national survey. We aimed to examine the neighborhood characteristics related to depressive symptoms in South Korean older adults and to suggest policy implications for neighborhood development. In addition, we examined the differences between depressive symptoms and regional characteristics of rural and urban areas.

All objective neighborhood characteristics were related to the depressive symptoms of older adults living in rural areas while only nursing homes were related to older adults living in urban areas. These results suggest that improving the objective neighborhood characteristics is more important to alleviate the depressive symptoms of older adults in rural areas. Notably, the number of social workers, senior centers, and home care services was related to lower depressive symptoms among older adults in rural areas. These findings suggest that expanding social services may improve older adults’ mental health. This result is consistent with the previous studies that social services can reduce depressive symptoms [13,37]. However, neighborhoods with a high number of nursing homes also had high levels of depressive symptoms in the entire sample. Older adults in nursing homes can experience social isolation and loneliness and are often more debilitated, which can contribute to depressive symptoms [38,39].

We found lower satisfaction with the perceived neighborhood and worse objective neighborhood characteristics in rural areas than in urban areas. The significantly lower satisfaction of perceived neighborhood characteristics among older adults living in rural areas means that older adults living in rural areas are exposed to inferior neighborhoods than urban areas. Rural areas had a higher proportion of older adults, a low region’s per capita income, and a low financial independence rate. This means rural areas were more economically vulnerable than urban areas, which is consistent with previous literature [4,21]. The multilevel models indicate that the socioeconomic vulnerability of neighborhoods is also related to the depressive symptoms of older adults. This result supports Wandersman and Nation [24]’s structural characteristics model that the structural characteristics of neighborhoods can have an individual psychological effect.

Older adults living in rural areas had higher average age, lower levels of education and income, poor self-rated health, and a higher likelihood of disability. Household size was small in rural areas, suggesting that older adults were more likely to be left alone in rural areas. We found that depressive symptoms were higher among those who were single, low-income, and with many comorbidities as well. This result is consistent with previous studies that depressive symptoms could be higher among those with no partner, low economic status, low education level, and chronic diseases [3,4,20,40,41,42]. These findings highlight that older adults who live in rural areas were more vulnerable and at increased risk of depressive symptoms.

The perceived neighborhood factors were significantly related to depressive symptoms even after applying for the individual characteristics. This result supports Lawton and Nahemow’s [12] assertion that health outcomes, including mental health, may differ depending on older adults’ interaction with the environment. Depressive symptoms were lower in areas where satisfaction with the current housing, the interaction of neighbors, and the overall community environment was desirable in the entire sample. This result was consistent with previous studies regarding housing, social network, and cohesion were related to depressive symptoms [41,43]. Policymakers should take these factors into account when considering improvements in neighborhoods. In addition, each municipality should examine whether the age-friendly elements (e.g., housing, transportation, outdoor space, buildings, etc.) are well-equipped in the neighborhood and provide support to improve the areas. The more older adults were satisfied with public transportation, the more they were likely to be depressed in this study. This result contradicts previous studies which found that better public transportation was associated with lower depressive symptoms [13]. We need future studies regarding this result.

### Limitations

The limitations of this study are as follows. First, since this study used cross-sectional data, it is difficult to determine the causal relationship between the neighborhood and depressive symptoms. The survey was conducted in the early stages of the pandemic, before vaccine availability. All of the variables, with the exception of depression, were based on 2019 data. We were unable to assess the COVID effect on older adults’ depression. However, it is possible that the pandemic had an effect on participants’ depressive symptoms. Based on the previous research (Kim, 2021) which was conducted in 2020, older adults living in rural areas had higher depressive symptoms during COVID-19 due to reduced social contact and worse health status. Future research needs to consider changes in older adults’ depression before and after COVID-19 using time series analysis or longitudinal analysis. Generally, transportation is negatively known to be related to depressive symptoms. However, the outcome of this study showed a different result from previous studies. Future studies need to identify causal relationships through a longitudinal study. Second, due to the limitations of data information, 8 metropolitan cities and 9 provinces were used as neighborhood levels. However, this is rather broad to set as a neighborhood. Future studies require analysis using smaller geographic criteria for finding neighborhood effects more accurately. Third, we chose objective neighborhood factors by focusing on social services and facilities in this study. Future research must examine the relationship between other factors such as mental care services or professionals that can affect mental health.

## 5. Conclusions

This study found that older adults who live in rural areas are more likely to experience depressive symptoms. Objective neighborhood characteristics are more important to older adults’ depressive symptoms in rural areas than those in urban areas. Particularly, these results suggest that expanding social services (i.e., senior centers, social workers, and home care services) may improve older adults’ mental health in rural areas. Perceived neighborhood characteristics also show lower satisfaction in rural as opposed to urban areas. Thus, policymakers should pay attention to developing rural areas to reduce older adults’ depressive symptoms. Although this study has several limitations, it is meaningful in finding the relationship between the perceived and objective neighborhood characteristics and depressive symptoms among older adults in South Korea after controlling the individual factors. This study can be used as basic data for proposing policies for older adults for the improvement of rural and urban environments.

## Figures and Tables

**Figure 1 healthcare-11-00476-f001:**
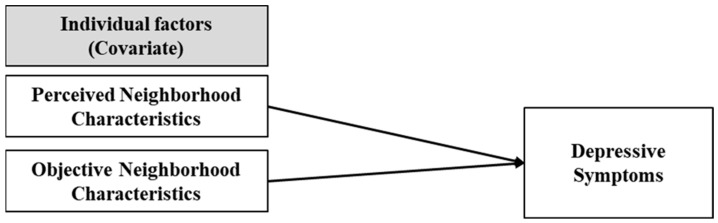
Conceptual Model.

**Table 1 healthcare-11-00476-t001:** Descriptive Findings by Region.

	Entire Sample (N = 10,097)		Region	
	M (%)	SD	Rural (N = 2867)	Urban (N = 7230)
**Depressive Symptom** ***	3.4	3.4	3.6	3.3
**Individual Factors**				
Age ***	73.6	6.3	74.7	73.1
Gender (female)	60.0	-	59.4	60.3
Marital status (couple)	58.7	-	59.2	58.6
Household size ***	1.9	0.8	1.8	1.9
Education (1–7) ***	3.8	1.2	3.5	3.9
Income (ten thousand won) ***	1489.3	2326.3	1227.0	1593.3
Employed ***	35.9	-	43.9	32.7
Self-rated health (1–5) ***	3.3	0.9	3.3	3.4
Disability (yes) *	4.5	-	5.3	4.3
Chronic disease	1.9	1.5	1.9	1.8
**Perceived Neighborhood (1–5)**				
Housing ***	3.8	0.7	3.7	3.8
Distance from the facilities ***	3.7	0.8	3.4	3.8
Public transportation ***	3.7	0.8	3.4	3.9
Green space sufficiency	3.8	0.9	3.7	3.8
Public safety ***	3.7	0.8	3.6	3.8
Distance from the family members ***	3.5	0.9	3.4	3.6
Interaction with neighbors **	3.8	0.7	3.7	3.8
Overall community environment ***	3.7	0.7	3.6	3.7
**Objective Neighborhood ^1^**				
Public social work administrator ***	3.2	0.0	3.3	3.1
Social worker ***	75.2	0.2	71.4	76.7
Senior center	3.3	0.5	4.6	2.7
Home care service	0.6	0.0	0.6	0.6
Nursing home ***	0.7	0.0	0.8	0.6
The proportion of older adults ***	16.2	3.3	18.5	15.3
Region’s per capita income ***	35,795.9	70.0	34,652.9	36,249.2
Financial independence rate ***	43.5	0.3	30.0	48.9

Note. **^1^** The number of facilities, workers, and services per 1000 older adults in each area * *p* < 0.05, ** *p* < 0.01, *** *p* < 0.001.

**Table 2 healthcare-11-00476-t002:** Multilevel Models.

	Model 1		Model 2		Model 3		Model 4a (Rural)		Model 4b (Urban)	
	b	SE	b	SE	b	SE	b	SE	b	SE
**Individual Factors**										
Age	−0.00	0.00	0.00	0.00	0.00	0.00	0.00 **	0.00	−0.00	0.00
Gender	−0.01 **	0.00	−0.02 *	0.00	−0.01 *	0.00	−0.02	0.01	−0.01	0.01
Marital status	−0.03 ***	0.01	−0.03 ***	0.01	−0.03 ***	0.01	−0.05 ***	0.01	−0.02 ***	0.01
Family members	−0.00	0.00	−0.00	0.00	−0.00	0.00	0.02 *	0.01	0.00	0.00
Education (1–7)	−0.01 ***	0.00	−0.01 ***	0.00	−0.01 ***	0.00	−0.00	0.00	0.01 ***	0.00
Income (ten thousand won)	−3.391 × 10^−6^ ***	8.97 × 10^−7^	−2.307 × 10^−6^ **	8.85 × 10^−7^	−2.305 × 10^−6^ **	8.85 × 10^−7^	−6.501 × 10^−6^ *	3.21 × 10^−6^	−1.78 × 10^−6^ *	8.89 × 10^−7^
Employment	−0.02 ***	0.00	−0.02 ***	0.00	−0.02 ***	0.00	−0.03 ***	0.01	−0.02 **	0.01
Self-rated health (1–5)	−0.08 ***	0.00	−0.07 ***	0.00	−0.07 ***	0.00	−0.07 ***	0.01	−0.07 ***	0.00
Disability (yes)	0.05 ***	0.01	0.05 ***	0.01	0.05 ***	0.01	0.06 ***	0.02	0.04 **	0.01
Chronic disease	0.02 ***	0.00	0.02 ***	0.00	0.02 ***	0.00	0.02 ***	0.00	0.02 ***	0.00
**Perceived Neighborhood (1–5)**										
Housing			−0.04 ***	0.00	−0.04 ***	0.00	−0.04 ***	0.01	−0.04 ***	0.00
Distance from the facilities			0.00	0.00	0.00	0.00	0.01 *	0.01	−0.01	0.00
Public transportation			0.01 **	0.00	0.01 **	0.00	0.01 *	0.01	0.00	0.00
Green space sufficiency			−0.00	0.00	−0.00	0.00	−0.01	0.01	0.00	0.00
Public safety			−0.00	0.00	−0.00	0.00	0.01	0.01	−0.00	0.00
Distance from the family			−0.00	0.00	−0.00	0.00	−0.00	0.01	0.00	0.00
Interaction with neighbors			−0.02 ***	0.00	−0.02 ***	0.00	−0.01	0.01	−0.02 ***	0.00
Overall community environment			−0.02 ***	0.04	−0.02 ***	0.00	−0.01	0.01	−0.02 ***	0.00
**Objective Neighborhood**										
Public social work administrator					0.02	0.02	3.10 ***	0.76	−0.04	0.03
Social worker					−0.00	0.00	−0.03 **	0.01	−0.00	0.00
Senior center					0.01 **	0.00	−0.45 ***	0.12	0.01	0.01
Nursing home					0.09 **	0.03	−3.30 ***	0.81	0.09 *	0.04
Home care service					0.06	0.03	0.68 ***	0.16	0.06	0.04
The proportion of older adults					−0.00	0.01	0.07 **	0.02	−0.01	0.01
Region’s per capita income					−1.11 × 10^−7^	1.25 × 10^−6^	−0.00 ***	0.00	−3.35 × 10^−7^	1.45 × 10^−6^
Financial independence rate					0.00	0.00	0.14 ***	0.04	0.00	0.00
Intercept	0.55 ***		0.74 ***		0.72 ***.		−7.14 ***		−1.70 ***	
**Model Statistics**										
AIC	−4118.98		−4457.98		−4457.56		−697.33		−3900.94	
BIC	−4025.35		−4025.35		−4208.69-		−524.90		−3701.79	

* *p* < 0.05, ** *p* < 0.01, *** *p* < 0.001.

## Data Availability

Data will be made available on request.

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
