# Peer review of "Neighborhood and Depressive Symptoms in Older Adults Living in Rural and Urban Regions in South Korea"

_healthcare, 2023, doi:10.3390/healthcare11040476_

Round 1
Reviewer 1 Report
This paper is of great interest as it depends on the relationship between depressive symptoms in older adults, neighborhood characteristics, and living in a rural or urban area.
The introduction perfectly situates the topic of study. I only miss some statistical data in this section about the population of older adults in South Korea.
The methodology is adequate to achieve the proposed objectives and is well presented in the manuscript.
The results are presented clearly.
In the discussion, the results are contrasted with other studies.
The conclusions are well defined.
Reviewer 2 Report
1. Will the impact of COVID-19 affect depressive symptoms in the samples because the data was conducted in 2020? If yes, the author(s) have to mention its impact in the Introduction and discussion section.
2. During the pandemic, will this affect some of the factors, including individuals (e.g., self-rated health), perceived neighborhood (e.g., interaction with neighbors, overall community environment), and objective neighborhood (e.g., public social work administration, social worker). Perhaps the author(s) may elaborate more in the discussion section.
Reviewer 3 Report
The authors conducted a cross-sectional observational study to examine the associations between neighborhood characteristics and depressive symptoms in older adults in South Korea. By analyzing the National survey data of 10,097 adults aged 65 and older, the authors showed that specific perceived and objective neighborhood characteristics were associated with depressive symptoms in older adults living in urban and rural areas. These findings are essential.
There are some comments.
Comments:
1. Methods (Data and Sample, Line 115 on Page 3): “We used the fifth survey conducted in 2020 with a national sample of 10,097 older adults aged 65 and over -.” How many adults aged 65 and over received the fifth survey? Please explicitly provide this information before this sentence. Also, were there eligibility criteria? Please describe them, if any.
2. Methods (Analysis plan): Multilevel modeling was performed in this study. The authors described how they built the models sequentially (Lines 188-197 on Page 5). In principle, multilevel modeling is used when the data have a grouping structure (for instance, multiple levels of nested groups) and when nested random effects are needed to be modeled. The random effects and the levels have to be specified in the analysis. However, it is unclear, based on the authors’ description, what the random effects were (for instance, random intercept and/or random coefficients), what levels the random effect was at (for instance, region), and what the assumed covariance structure for the random effects was. A more detailed description is necessary.
3. Results (Table 1): Individual and neighborhood characteristics of older adults living in rural and rural regions were presented. Please provide the number and percentage for categorical variables. Please also give the standard deviations for continuous variables.
4. Results (Lines 276-278 on Page 7): “- as age and household size increased, the depressive symptoms decreased statistically in the rural areas.” “The higher education level significantly reduced depressive symptoms in urban areas -.” These descriptions contradict the results shown in Table 2. Please recheck the description.
5. Discussion (Line 290 on Page 8): “- while only nursing homes were related to older adults living in rural areas.” This description contradicts the results shown in Table 2. Please recheck the description.
6. Discussion (Line 297 on Page 8): “neighborhoods with a high number of nursing homes also had high levels of depressive symptoms in both areas.” This description contradicts the results shown in Table 2. Please recheck the description.
7. Abstract (Line 20 on Page 1): “only nursing homes (b=.91, p<.05) were related to depressive symptoms of older adults living in urban areas.” This description contradicts the results shown in Table 2. Please recheck the description.
8. Title: Please delete “A Comparison of.” The title could be revised as “Neighborhood and Depressive Symptoms in Older Adults living in Rural and Urban Regions in South Korea.”
